# Motivation, barriers and preferences of lifestyle changes among older adults with frailty and mild cognitive impairments: A scoping review of qualitative analysis

Jamilah Mohammad Hanipah[1]☯, Arimi Fitri Mat Ludin[1,2]☯*, Devinder Kaur Ajit Singh[1,3]☯, Ponnusamy Subramaniam[1,4]☯, Suzana Shahar[1,5]☯

1 Centre for Healthy Ageing and Wellness (H-CARE), Faculty of Health Sciences, Universiti Kebangsaan Malaysia, Kuala Lumpur, Malaysia, 2 Programme of Biomedical Science & Faculty of Health Sciences, Universiti Kebangsaan Malaysia, Kuala Lumpur, Malaysia, 3 Physiotherapy Programme, Faculty of Health Sciences, Universiti Kebangsaan Malaysia, Kuala Lumpur, Malaysia, 4 Clinical Psychology and Behavioral Health Programme, Faculty of Health Sciences, Universiti Kebangsaan Malaysia, Kuala Lumpur, Malaysia, 5 Programme of Dietetic, Faculty of Health Sciences, Universiti Kebangsaan Malaysia, Kuala Lumpur, Malaysia

☯ These authors contributed equally to this work.
* arimifitri@ukm.edu.my

## Abstract

Lifestyle intervention has proven effective in managing older adults' frailty and mild cognitive impairment issues. What remains unclear is how best to encourage lifestyle changes among older adults with frailty and Mild Cognitive Impairment (MCI). We conducted searches in electronic literature searches such as PubMed, Scopus, Cochrane Reviews, ProQuest, and grey resources to find articles published in English between January 2010 and October 2023. This review focused on research using a qualitative study design. We extracted data on publication year, location, the aim of the study, study population, involved intervention, barriers, motivations, and preferences reported in the articles. Out of 5226 retrieved, 253 articles were selected after the deletion of duplicates, title, abstract screening, and. We included fourteen articles for final analysis at the end of the review process. The two main themes generated from this review are intrinsic and extrinsic factors in motivations and barriers to lifestyle changes. The most reported motivators were the perceived benefits of lifestyle intervention and self-efficacy. Among the obstacles participants face are perceived adverse effects of intervention, lack of knowledge, existing impairment (physical or mental), and social support. Lifestyle change motivations and barriers among older adults mainly were intrinsic factors such as the perceived benefit of the intervention, self-efficacy, knowledge, familial commitment, and existing impairments. There is a need to empower older adults to overcome the barriers with support from healthcare professionals, the community, and the family.

**Data Availability Statement:** All relevant data are within the article and its Supporting Information files.

**Funding:** This study is funded by the Ministry of Higher Education under the Long-Term Research Grant Scheme (LRGS/1/2019/UM-UKM/1/4)) however, the funder funders had no role in study design, data collection and analysis, decision to publish, or preparation of the manuscript. This study also has been approved by the Research Ethics Committee National University of Malaysia (UKM/PPI/111/8/JEP-2020-34).

**Competing interests:** The authors have declared that no competing interests exist.

## Introduction

Ageing is [1] associated with an increased risk of chronic diseases, cognitive decline, and physical frailty, all of which significantly impact the quality of life of older adults [2, 3]. Frailty and cognitive impairment are two of the most prevalent and challenging conditions in geriatric populations, with both often occurring simultaneously [4]. Cognitive frailty (CF) is a relatively new clinical concept, defined as co-occurrence of physical frailty and mild cognitive impairment (MCI) in older adults without dementia [5]. This simultaneous presence of frailty and cognitive decline places older adults at higher risk for adverse outcomes, including functional dependency, institutionalisation and mortality, compared to those who experience either condition alone [6–8]. Unlike dementia, CF is potentially reversible if addressed early with appropriate interventions [9].

Lifestyle intervention or modification has strong evidence-based effects on older adults with physical frailty and cognitive impairment [10–12]. Lifestyle interventions had positive outcomes in reversing or delaying the progression of both physical frailty and mild cognitive impairment [13–15]. This intervention includes multiple domains approaches such as physical (exercise and physical activity), nutrition (healthy and balanced diet with supplements), cognitive (cognitive stimulation activities), psychosocial (social support), as well as cardiovascular risk management (cardiovascular health, control of smoking/ alcohol consumption), is also successful in addressing [16–18] and preventing adverse health outcomes [19]. However, participation in these interventions is influenced by various intrinsic and extrinsic barriers [20]. Understanding these barriers, alongside the motivators that encourage participation, is essential in designing effective and accessible intervention programs.

Although there are recent CF-related studies, it is confined to the effectiveness of various interventions, risk factors, predictors, adverse effects, related biomarkers, and reversibility of the condition [7, 11, 21–24], rather than on factors that influence older adults' willingness and ability to participate in such interventions. Therefore, this review includes research conducted on older adults with physical frailty or MCI, as these populations are critical to understanding cognitive frailty's two key components. By examining studies that address both physical frailty and MCI, this review provides a foundation for understanding how to increase participation in lifestyle interventions tailored to older adults with cognitive frailty. Translating and implementing evidence-based interventions into healthcare settings necessitates understanding population-specific motivations and barriers at the individual, organizational, and community levels [25].

Therefore, in this review, we aim to identify the available evidence on the perspectives of older adults with frailty and MCI towards lifestyle interventions or modifications. Specifically, this scoping review aims to identify and map the motivations, barriers, and preferences toward lifestyle intervention/modification among older adults with frailty and MCI. This study is part of an AGELESS Trial study under the LRGS scheme aimed at identifying an effective intervention for the reversal of cognitive frailty for older adults in Malaysia [26].

## Methodology

The method of this scoping review is based on the framework developed by Arksey and O'Malley [27] and Joanna Briggs Institute's updated framework [28]. In addition, we utilized the approach suggested by Levac, Colquhoun, and O'Brien for searching, screening, and reporting of scoping review was used for this review [29]. These six stages in conducting a scoping review, include (1) Identifying the research question; (2) relevant studies; (3) study selection; (4) charting the data; (5) collating, summarizing, and reporting the results; (6) consultation with stakeholders. Reporting of this scoping review findings will be based on Preferred Reporting Items for Systematic Reviews and Meta-Analyses extension for Scoping Reviews (PRISMA-ScR) Checklist [30].

**Table 1. Table of population, concept and context that is used as a guide to formulate keyword and search strings.**

| | |
|---|---|
| Population | Older adults (Aged 60 and above, community-dwelling, having MCI or physical frailty) |
| Concept | Outcome: Perception towards lifestyle changes or intervention programs (physical activity, healthy diet, cognitive training, and psychosocial activities) |
| Context | Motivation, barriers, and preference toward lifestyle changes |

## Stage 1: Identifying the research question

In our review, we aim to answer these questions.

What are the motivations among older adults with frailty and MCI to participate in lifestyle intervention/programs?

What are the barriers among older adults with frailty and MCI to participate in lifestyle intervention/programs?

What is lifestyle intervention/program preferences among older adults with frailty and MCI?

## Stage 2: Identifying relevant studies

A comprehensive search strategy was discussed and developed among the team members. We used PCC (Population, Concept, and Context), as shown in Table 1 below as a guide. Keywords/search terms were extracted from the title and main objective. Synonyms for the keywords were then derived using Medical Subject Heading (MeSH) and online thesaurus search.

Search strings (Table 2) were developed using the Boolean Operator to combine the synonyms. We searched through electronic databases (PubMed, Scopus, Cochrane Reviews, and ProQuest) as well as grey literature on Google Scholar. Searches were limited to the studies published between January 2010 and October 2023, full-text articles, human subjects, and in

**Table 2. List of search string generated from keywords and its synonyms.**

| | |
|---|---|
| Motivation | ("Motivation" OR "Interest" OR "Facilitate*" OR "Enabler" OR "Desire" OR "Reason" OR "Adherence" OR "Compliance" OR "Participation") |
| Barrier | ("Barrier" OR "Limit" OR "Obstacle" OR "Difficulty" OR "Restriction" OR "Drawback" OR "Dropouts") |
| Preference | "Preference*" OR "Choice*" OR "Option*" OR "Favourite" OR "Selection" OR "Inclination" OR "Like*" |
| Lifestyle | ("Lifestyle changes" OR "Lifestyle intervention" OR "Healthy lifestyle" OR "Lifestyle pattern" OR "Healthy behaviour" OR "Lifestyle modification" OR "self-management" OR "Lifestyle behaviour") |
| Exercise | ("Exercise" OR "Physical exercise" OR "Exercise training" OR "Exercise therapy" OR "Exercise participation" OR "Physical activity") |
| Psychosocial intervention | ("Psychosocial intervention" OR "Psychosocial approach" OR "Psychosocial support") |
| Cognitive training | ("Cognitive training" OR "Cognitive stimulation" OR "Brain exercise" OR "Brain game") |
| Nutrition intervention | ("Nutrition intervention" OR "Nutrition Enrichment" OR "Healthy nutrition" OR "Healthy diet" OR "Healthy eating" OR "Healthy meal") |
| Older adults | ("Older adults" OR "Elderly" OR "Geriatric" OR "Aged" OR "Older population" OR "Aging population") |
| Mild Cognitive Impairment | ("Mild cognitive impairment" OR "Cognitive decline" OR "Cognitive impairment" OR "cognitive dysfunction") |
| Physical Frailty | ("Physical frailty" OR "Frailty" OR "Physical impairment" OR "Physical decline" OR "Physical weakness" OR "Weakness") |

English. Hand-searched articles were also searched using the same search criteria. The most recent search was executed on 1st November 2023. All retrieved articles were uploaded to Mendeley's application, merged into one shared folder and screened for duplication.

## Stage 3: Selection of relevant studies

Articles selection was based on the inclusion criteria: (1) studies that report on motivation, barriers, or preferences of older adults with frailty or MCI towards lifestyle modification or participation in a lifestyle intervention program, (2) participants aged 60 and above, (3) study design was qualitative and mixed method, (4) published in the English, (5) studies published between January 2010 and October 2023. Review articles were excluded from this scoping review.

Lifestyle interventions in this review are defined as structured programs aimed at promoting health related behavior changes across multiple domains [31]. These interventions include. physical activity (PA), nutritional or dietary modification, cognitive training, and psychosocial intervention. The term "physical activity" is in this review reflect the broad range of definition and not limited to the "planned, structured, repetitive, and purposive" activities [32]. Nutritional intervention includes planned actions to positively change nutrition-related behavior, risk factors, environmental condition, or health status for an individual, family, caregivers, target groups, or the community [33]. Cognitive training refers to cognitive-based guided practice and structured tasks incorporated into interventions to improve cognitive function [34]. Finally, psychosocial intervention is called ANY non-pharmacological intervention that aims to change cognitive function and improve an individual's health symptoms, functioning, and well-being [35].

The screening process consists of two stages: title screening and abstract screening. The online systematic article management software Rayyan Ai was used to facilitate the screening process [36]. Rayyan AI software was employed to streamline the organization and screening process to manage duplicates and facilitate title and abstract screening. In the first stage of the literature search, we found 5042 articles and included 245 articles from reference tracking. Next, JMH and AFML independently screened the title and abstract of the articles. At the end of stage one, 5143 articles were excluded. JHM and AFML screened 144 full articles, then read the full articles in detail to determine relevance of individual articles. Any discrepancies were resolved during regular census meetings among all the authors. Fig 1 shows the flow diagram of search and study selection, adapted from the PRISMA group [30]. A total of 14 articles were included in the final analysis [21, 37–49].

## Stage 4: Charting the information from the selected studies

The first and second authors independently read the included articles and extracted any motivation, barriers, or preferences identified in each paper based on the definition above. Following the extraction of data, the studies were categorized and tabulated based on the following details:

a.  Authors, publication year, and country.

b.  Objective or purpose of the study

c.  Participant's characteristics (–age, sociodemographic factors, frailty/cognitive status)

d.  Study design

e.  Type of intervention/program used or tested, and type of outcome measured.

f.  Motivation, barriers, or preferences reported.

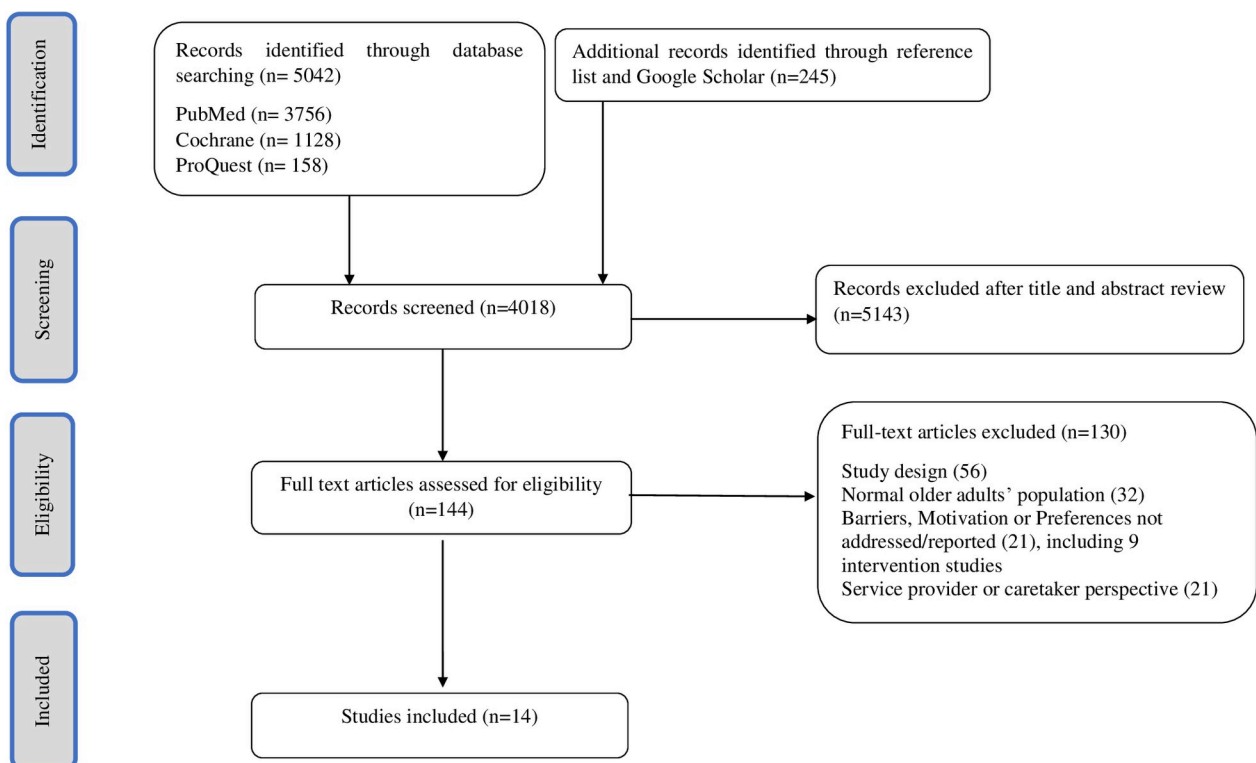

**Fig 1. Preferred Reporting Items for Systematic Reviews and Meta-Analyses Extension for Scoping Review (PRISMA-ScR) flow diagram for motivation, barriers, and preferences among older adults with frailty and mild cognitive impairment towards lifestyle changes.**

"Barriers" were defined as any physiological, psychological, or socio-ecological conditions reported to reduce or negatively affect a person's participation in any lifestyle intervention. "Motivation" is defined as a comprehensive term that includes both internal motivators (such as personal goals, perceived benefits and psychological readiness) and external factors (such as social support, accessibility and environmental conditions [50–52]. Preferences were defined as characteristics or features of any lifestyle interventions participants identified as enjoying. The final consensus to include in the analysis was achieved during the regular meeting among all the authors (SS, DKAS, or PS).

## Stage 5: Reporting of results

The selected articles were then stored in NVivo 12 Plus, qualitative data analysis software, for extraction and thematic data analysis [53]. First, JMH examined the data and familiarized itself with it to understand and facilitate generating the initial codes. The initial codes were formed, and the quotes were categorized according to the appropriate theme. Next, JMH classified the themes to find the relationship between the themes. The categorized themes were classified and named upon discussion with AFML. The identified themes were finalized during the census meeting with all the authors.

## Results

### Study characteristic

Of 5287 unique references identified through searches, 14 studies were included in this review (Fig 1), of which explored participants' perceptions and experiences regarding lifestyle

interventions or modifications. These studies examined various aspects of lifestyle interventions, including participants' views on the feasibility, acceptability, and perceived effectiveness of programs aimed at improving physical or cognitive health outcomes. Two studies reported on the perceptions of older adults with MCI towards PA [42, 43]. One study reported the barriers and preferences of older adults with MCI in participation in computer-based cognitive training [37]. One study reported on the motivation and barriers of older adults with MCI towards digitalised multicomponent lifestyle intervention programs [45]. Another selected study reported barriers and preferences of older adults with MCI towards psychosocial approach (public square dance) [44], and two studies reported on motivation, barriers, and preferences of older adults with MCI towards multi-component lifestyle interventions to prevent cognitive decline and cardiovascular risk factors [21, 46]. For the participants with frailty, three of the selected studies reported on the motivation, barriers, and preferences of these populations toward PA [38, 40, 47]. Four studies selected from participants with frailty reported on older adults' perceptions of multi-component interventions [39, 41, 48, 49]. Characteristics of the selected studies are tabulated in Table 3.

In this review, we identified a few overarching themes that included both intrinsic and extrinsic factors. Like Mohamed Nor et al. [54], we categorize the motivations and barriers into intrinsic and extrinsic categories as the central theme. The sub-themes from the primary themes were classified as either motivations or barriers to participation in lifestyle interventions.

**1. Motivations to participate in lifestyle changes.** The motivations identified under intrinsic factors were the perceived benefit of the intervention, self-efficacy, knowledge and diagnosis of the condition. The motivations explored under extrinsic factors were the source of information, social support, intervention-related component, and accessibility. Table 4 shows the summary of themes for motivations factors.

*1.1 Intrinsic motivators to participate in the lifestyle intervention. a. Perceived benefits of lifestyle intervention*: Ten articles highlighted the benefits of lifestyle interventions as motivations for participation to make changes [38, 40–43]. Participants experienced physical and mental health benefits from engaging in interventions [43]. Maintaining physical status reduced the burden and stress for care partner and family [40, 42, 46]. Social engagement during interventions was also a perceived benefit [40, 41, 43, 46].

*b. Self-efficacy*: Self-efficacy motivates participation in lifestyle interventions. Self-efficacy is crucial in implementing lifestyle changes as it reflects the willpower to either start or not start the lifestyle changes [55]. Participants from one of the studies pointed out that those behaviors associated with independence and a sense of self-development over a lifetime are motivations for them to make lifestyle changes. They also acknowledge that the behavior changes benefits brain health [45]. Positive mood helps them engage in lifestyle changes intervention program [43]. Participants with MCI desired to maintain or increase their current physical status, which determined their engagement in PA. Self-efficacy qualities of strong internal motivation and discipline to exercise, time management skills, and the ability to adapt to changing schedules determine engagement in intervention [42].

*c. Knowledge*: Understanding the disease helps older adults shape their attitudes toward cognitive disorders and their prevention and moderates their expectations and reasons for participation in prevention trials [21]. Knowledge of local services and the skills to access them (e.g., form filling) also motivated older adults to participate or make lifestyle changes [38]. Additionally, participants' intention to contribute to the scientific knowledge of MCI and its prevention also is a reported motivation for their participation in such activities [46].

*d. Diagnosis of condition*: Participants in one study mentioned that the diagnosis of MCI or early dementia was motivation for exercising more [43]. Few studies suggest that family

**Table 3. Characteristics of selected studies.**

| | Author/Year/ Country | Aim | Participant's characteristic | Study design | Intervention |
|---|---|---|---|---|---|
| 1 | Hobson *et al.*, 2020 Canada | To understand the barriers, facilitators, and preferences for exercise among persons living with MCI or early dementia. And their care partners. | Community-dwelling older adults aged 55 and above living with MCI or early dementia. (n = 17) 10 persons living with MCI or early dementia (five with MCI and five with dementia) and seven care partners. | Qualitative study using semi-structured interviews. | Exercise. |
| 2 | Bechard *et al.*, 2020 Canada | To explore the perceptions, experiences, and beliefs of older adults with cognitive impairment and their caregivers concerning physical activity. | Community-dwelling adult aged ≥65 years diagnosed with MCI or mild-to-moderate AD and their care partner (10 caregivers and 10 participants with cognitive impairments) | Qualitative exploratory study, semi-structured interviews. | Physical activity (PA). |
| 3 | Åhlund *et al.*, 2020 Sweden | To explore the perceptions of physical activity and exercise among frail elderly patients with a severe comorbidity burden. | Community dwelling, frail older adults ages 75 years or older. Assessed as frail according to the FRESH screening instrument. (n = 18) | Qualitative study with semi-structured interviews | Physical activity and exercise. |
| 4 | Jadczak *et al.*, 2018 Australia | To explore pre-frail and frail older peoples' perspectives in relation to being advised about exercise and their perceptions of the general practitioners' (GPs) role in promoting exercise for older people. | Community-dwelling older adults aged 75 years and above are screened pre-frail or frail using the FRAIL Screen. (n = 12) | Qualitative study using semi-structured interviews. | Exercise |
| 5 | Yu *et al.*, 2020 Hong Kong | To explore pre-frail and frail older Chinese people's perspectives on a multi-component, group-based frailty prevention program in Hong Kong, along with their views regarding factors determining participation and sustainability of such a program. | Community-dwelling older people aged 54–84 screened with pre-frailty or frailty (n = 38) | Qualitative–focus group discussion. | 12-week multi-component (involving physical, cognitive, and social activities), group-based frailty prevention program |
| 6 | Anna Rosenberg *et al.*, 2020 Finland | To explore participants' knowledge and perception of cognitive impairment and dementia, as well as attitude towards prevention. | Community-dwelling older adults aged 65 and above. (n = 15) | Qualitative with semi-structured interviews. | Interactive stimulating self-management of vascular risk factor using internet platform. |
| 7 | Frost *et al.*, 2018 England | To explore perceptions of health promotion behaviors undertaken by older people with mild frailty, barriers, and facilitators to engage, and identify potential components for new home-based health promotion services. | Community- dwelling adults aged ≥75 years (n = 53) 14 mildly frail older people, 12 family carers, 19 community health and social care professionals, and 8 homecare workers, in one urban and one semi-rural area of England. | Qualitative study using semi-structure questionnaire | Home- based health promotion services. |
| 8 | Haesner *et al.*, 2015 Germany | To assess specific preferences and potential barriers of older adults regarding a web-based platform for cognitive training. | A total of 12 older adults aged over 60 (6 healthy and 6 older with MCI.) | Qualitative study (semi-structured interviews) | Computer based cognitive training |
| 9 | Yao *et al.*, 2021 Beijing | To explore the acceptability and feasibility of public square dancing among community residents with mild cognitive impairment (MCI) and depressive symptoms. | Community-dwelling older adults aged 60 to 85 years who complained of memory loss. Cognitive status assessed using Montreal Cognitive Assessment (MoCA-P) | Mixed-method study consisted of quantitative and qualitative phase. | Psychosocial intervention -Public square-dancing intervention |
| 10 | Burton *et al.*, 2022 Ireland | To explore the experiences of older adults undertaking the CTM approach within their home support services and to identify the strengths and barriers of adopting CTM from the perspective of the older recipient. | Older adults (≥65 years), in receipt of home care support, who had a Clinical Frailty Score of 6 or less, had fallen at least once in the last year, and were independently mobile (with or without a walking aid) (N = 13) | Qualitative semi structured interviews via–descriptive approach | "Care To Move" (CTM)–pre-set movement prompts that a deployed by workforce as part of everyday activities. |

(*Continued*)

**Table 3.** (Continued)

| | Author/Year/ Country | Aim | Participant's characteristic | Study design | Intervention |
|---|---|---|---|---|---|
| 11. | Canet-Vélez et al., 2023 Barcelona, Spain | To explore the perspective of frail older adults and professionals about the barriers, facilitators, and improvement elements of the development of the +AGIL Barcelona program including its digital component. | Individuals aged ≥80 years (n = 11), presenting at least one sign of frailty | Qualitative descriptive method–content analysis | +AGIL–multicomponent care intervention for frail older adults that involves primary, geriatric and community care. |
| 12. | Akenine et al., 2022 Sweden | To explore and describe the experiences of participation in the Multimodal preventive trial for Alzheimer's disease (MIND-AD MINI) among persons with prodromal AD. | Older adults aged 60–85 years, MMSE score of ≥24, symptoms of prodromal AD (N = 8) | Qualitative, Semi-structured face-to-face interviews | MIND-AD (MINI)– 6-month multidomain intervention with 3 parallel arm–(i. multidomain lifestyle/vascular intervention) (ii. multidomain lifestyle/vascular intervention + medical food). |
| 13. | Villa-García et al., 2023 Barcelona | Aims to identify implementation strategies for optimising the accessibility, acceptability and adaptability of mHealth interventions aimed at increasing physical activity. | Older adults presenting with at least one sign of frailty, with no or minimal disability in performing basic activity of daily living | Mixed-method study–adopting triangulation multilevel model. | "AGIL Barcelona (AGILBcn)" community-based integrated care program is a multicomponent healthy aging intervention for frail older adults—physical activity, nutrition, emotional wellbeing, sleep hygiene, cognitive screening and stimulation, loneliness, and medication review. |
| 14. | Essery et al., 2021 United Kingdom | Aims to explore whether a digital approach appears to be feasible, engaging and acceptable means of delivering a low-cost, multi-domain intervention to reduce cognitive decline among older adults. | Older adults age range between 61–80 recruited from GP clinics–physically inactive/ sedentary (Godin Leisure Time Exercise Questionnaire), no nondementia diagnosis, Scored as "Lower cognitive performance" on the Baddeley verbal reasoning task. | Mixed method study–qualitative Semi structured interview | 'Active Brains': a multi-domain digital behavior |

history or indirect experiences of cognitive disorders may motivate some older adults to seek medical advice and information about health and preventive measures. Fear and family history of cognitive disorders were mentioned as crucial factors toward lifestyle changes and

**Table 4. Summary of findings for motivations theme.**

| Main themes | Emerging sub-themes | Descriptions |
|---|---|---|
| Intrinsic motivations | Perceived benefit of the intervention | • For well-being<br>• Decrease caregiver stress.<br>• Meaningful activity<br>• Social engagement |
| | Self- efficacy | • Health behaviour<br>• Positive mood<br>• Proactive attitude towards intervention |
| | Knowledge | • Knowledge of disease<br>• Knowledge of available services |
| | Diagnosis of condition | • Diagnosis as motivators–self or family member<br>• Minimise burden on family |
| Extrinsic motivations | Source of information | • Health information, lifestyle changes advise and recommendation.<br>• Trustworthy source–sense of urgency |
| | Social support | • Family and member of community<br>• Service provider encouragement |
| | Accessibility | Environment<br>• Transport |
| | Intervention related | • Personalisation of intervention<br>• Service provider expertise and skill |

engagement in the prevention [45]. Those diagnosed with cognitive disorders feel motivated to participate as they want to minimize the burden on the family and carer [46]. Participants discussed their experiences with affected people are more likely to be interested in participating in lifestyle interventions [21].

*1.2 Extrinsic motivations to participate in lifestyle intervention. a. Source of information*: Recommendations from healthcare professionals [physicians or therapists] induce a sense of urgency and fear, motivating them to initiate and maintain exercise habits [43]. For many participants, knowledge or information from their providers seemed to be a necessary for them to have the confidence to make changes [42].

*b. Social support*: Seven of the included studies mentioned social support as the facilitator for participating in lifestyle interventions. Encouragement and support from others were found to motivate exercise and influence health behavior [43, 45–47]. Care partners enable participants to engage in PA and greatly impact participation among the MCI population [42]. A social support network provides support, freedom and opportunities to socialize and contribute to the community [39]. Creating action plans with others, being accountable. And sharing encouragement and new ideas with others also motivates participation in lifestyle changes [45].

*c. Accessibility*: Accessibility to the intervention program was one of the motivating factors for participants with frailty to participate and adhere to lifestyle intervention. Proximity to community resources, access to exercise equipment, and supportive care were physical environment that supported and facilitated the participants [42]. In addition, the availability of assistive devices to facilitate PA also enhance their accessibility to the intervention program [40].

**2. Barriers to participating in lifestyle changes.** There were more barriers to initiating or maintaining lifestyle changes than motivations. These barriers can further be categorized into intrinsic and extrinsic factors. The intrinsic barriers identified were perceived negative experience of the intervention, lack of knowledge, family commitment, and lack of intrinsic motivation. The extrinsic barriers explored were care partner support and availability, accessibility, social stigma, and intervention-related (Table 5).

*2.1 Intrinsic barriers to participating in lifestyle interventions. a. Perceived negatively experienced intervention*: Participants identified negative experiences during PA (e.g., disliking the feeling of sweating or being in the water while swimming), following PA (e.g., sore or aching muscles, fatigue), as a significant barrier to engaging in interventions [43]. Another study reported participants' lack of understanding of the benefits of exercise, perceiving it as challenging and potentially ineffective for older adults [40]. Experience and discrepancy between their experiences and what they had heard about prevention were also mentioned as barriers to participating in lifestyle interventions [21].

*b. Lack of knowledge*: Lack of knowledge act as a barrier to lifestyle intervention participation, hindering participations' ability to perform activities and access relevant information [40]. In programs utilizing digital technologies, participants may lose interest due to insufficient skills with devices [46, 49]. Participants with MCI may feel fearful and ashamed of their diagnosis, leading to reluctance in learning about their condition [21].

*c. Family commitment*: Family commitment as a barrier was reported mainly by female participants. Their responsibility towards their family members, where their role as a carer may override their self-care, hobbies, and other activities [38, 49].

*d. Physical and memory impairment*: Participants stated their physical impairments [mobility restrictions, pain and arthritis, visual impairment, and other physical changes] limit their ability to exercising, particularly walking [45, 48]. Memory issues and confusion also hindered

**Table 5. Summary of findings for barriers theme.**

| Main themes | Emerging sub-themes | Descriptions |
|---|---|---|
| Intrinsic barriers | Perceived negative experience of the intervention | • After effect of exercise |
| | Lack of knowledge | • Does not understand the benefit.<br>• Does not know how to perform.<br>• Reluctance to learn about own disease process. |
| | Family commitment | • Female especially |
| | Physical and mental impairment | • Memory problem- unable to remember schedule or steps.<br>• Increase dependency to caretaker.<br>• Pain and physical disability |
| | Lack of intrinsic motivation | • Lack of confidence<br>• Lack of interest<br>• Lack of motivation to perform without supervision |
| Extrinsic barriers | Care partner support and availability | • Availability–to participate together as they need help or guidance to handle the intervention or in terms of transportation.<br>• Health status of carer |
| | Accessibility | • Environment and infrastructure<br>• Transportation–not qualified for community assistance as MCI or dementia not considered as disability.<br>• Availability of specific program<br>• Climate change |
| | Social stigma | • Perception by others<br>• Denial of expected situations |
| | Intervention related | • Task too demanding<br>• Not personalised based on their level or ability<br>• Service provider continuity<br>• Lack of sufficient number of service provider. |
| | Information | • Does not get the information from their healthcare providers |

participation in the intervention, especially exercise at home [43]. Physical impairments were consistently identified as barriers to participation [38].

*e. Lack of intrinsic motivation*: Motivation plays a crucial role in lifestyle changes. With motivation, most participants were able to make the necessary changes. Participants who lacked motivation struggled to make necessary changes, such as exercising more [43]. Some perceived as too old to walk or exercise, dampening their interest in participation [47]. Frail older adults expressed a lack of motivation to exercise alone at without supervision, impacting program [49].

*2.2 Extrinsic barriers to participating in lifestyle intervention. a. Care partner support and availability*: Due to aging and disease processes, individuals with cognitive decline may rely more on their care partners for companionship and assistance with exercise. In addition, participants need their carers to co-participate as they need help or guidance during the intervention or in terms of transportation. However, care partners may not always be available due to other commitments. The health status of caregivers may also impact their ability to participate in the program [42, 43].

*b. Accessibility*: Three selected studies mentioned accessibility as a barrier to participating in lifestyle interventions. Different aspect of accessibility was mentioned such as accessibility in terms of environment, transportation, availability of specific programs, and climate change. Transportation can be a significant barrier for older adults to access intervention or program locations as some cannot drive or take public transport [43]. Another aspect of barriers related to accessibility is the availability of specific programs to their mental or physical status. Most programs required a care partner to be present, which was impossible for all persons living

with MCI or early dementia [42]. Persons living with MCI or early dementia did not qualify for community-assistive transport for persons with disabilities as MCI or early dementia was not considered a disability [38].

*c. Social stigma*: Perceived societal stigma was stated to have a negative influence on exercise participation. They felt that people perceived them as diseased instead of individuals [43]. Lack of understanding from peers and exercise professionals about cognitive impairment also created uncomfortable situations deterring some participants from PA participation [43].

*d. Intervention related aspects*: Participants in one of the studies stated that the task given in the computerized cognitive training was too complicated, and it demotivated them to continue their participation. When a task is too demanding for individuals with MCI, they could be reluctant to communicate online with strangers [37]. In addition, an intervention not tailored to a participant's ability and condition tends to demotivate them from participating [21]. On the other hand, service providers factor was also mentioned in one of the studies where the service provider (therapist/trainer) continuity does affect their motivation to continue engagement with intervention programs. Frequent change or use of locum staff indirectly affects the interpersonal relationship between participants and service providers and demotivate them to continue with the program [47].

**3. Preferences towards lifestyle interventions.**   We identified two sub-themes emerging from the main theme preferences (Table 6). First, preferences refer to characteristics or features of any lifestyle interventions participants identified as enjoying or encouraging.

*3.1 Intervention-specific preferences*. Seven of the selected studies reported participants' preferences for different lifestyle interventions. Participants preferred a personalized group exercise program with individuals with similar issues. Such engagement induced feelings of comfort and enjoyment and a sense of self-worth [43]. Participants also shared that the intervention needs to be tailored to their physical and mental needs to be more fun and enjoyable [40, 43, 47]. Participants in another study preferred the interventions conducted in different groups based on their ability level [41]. In a study on the perspective of older adults toward computer-based cognitive training, participants expressed that they would prefer regular

**Table 6.  Summary of findings for preferences theme.**

| Main themes | Emerging sub-themes | Descriptions |
|---|---|---|
| Preferences | Intervention specific | • Personalised program specific to their need and ability<br>• Intervention component–progressive, feedback system to inform and motivate participants.<br>• Incorporate daily activities into the intervention.<br>• Social engagement component–grouped activities, within familiar community or similar group (MCI or frailty), face to face interaction.<br>• Training environment–easily accessible, within community or transport provided.<br>• Intervention provider/ Instructors–knowledgeable, well trained, friendly and able to engage with participants.<br>• Someone who can develop good relationship–trustworthiness–increase adherence.<br>• To be organised by church, governmental agencies or medical service providers.<br>• Intervention cost–free or at lower cost<br>• Increase the duration of program/supervision–to avoid losing the benefits gained. |
|  | Information | • Promote participants knowledge on the health issues, available intervention, and services.<br>• Written information preferred–memory problem.<br>• Shared responsibility between participants and health service provider. |

training sessions specifically designed to improve cognitive abilities as opposed to sporadic ones [37].

Participants stated that they would like a progressive intervention program with a feedback system to inform their progression and performance to motivate them [37]. Participants also proposed to have shorter initial assessments and encourage feedback sessions between service providers and participants to improve the coordination and understanding of their progress [49].

In addition, the intervention program (computer-based cognitive training) should incorporate daily activities. Activities done in small groups based on a variety of light, but challenging exercises are preferred by the participants [43, 46, 48]. Other than that, participants preferred if the training location was accessible within the community or provided transportation [38].

Furthermore, participants from one of the studies share that they prefer face-to-face intervention/program as compared to online mode of delivery. Face-to-face mode of delivery is preferred as they can have social interaction and support during the intervention program [48]. Participants also proposed to increase the program duration to avoid losing the gained benefit [49].

Preference for the intervention was also mentioned about the intervention providers. Participants preferred knowledgeable, friendly instructors who could engage with participants [39, 47]. This will indirectly improve adherence among older adults, especially with MCI or frailty. It was also suggested that religious, governmental agencies or medical service providers organize the intervention. Some participants expressed their wishes for the continuity of the program at an affordable price. Providing free or low-cost affordable frailty prevention programs would be an essential strategy for frailty prevention [21, 37, 38, 44].

*3.2 Information.* Participants in one of the studies expressed that the local government should be more engaged in promoting PA programs for older people [40]. In addition, promoting increased knowledge through information should be a shared responsibility between healthcare providers and clients. Participants also preferred brochures with information, including the health benefits of exercise in older age and available PA programs in the community [43]. Participants prefer the intervention related information to be shared in written format as they feel it is easier for them to understand and in a way is helpful for them to remember [46].

## Discussion

This review evaluated the recent literature on motivation, barriers, and preferences toward lifestyle intervention among older adults with MCI or frailty.

We found that the intrinsic factors were a more significant concern compared to extrinsic factors. In addition, the results indicate that participants reported more barriers than motivations in lifestyle changes. The most reported sub-themes under barriers are the perceived negative effect of the intervention and lack of knowledge. On the other hand, perceived benefits of the intervention were the most reported sub-themes among participants in the motivation factor theme. In addition, adequate knowledge, social support, and motivation facilitated engagement in lifestyle changes.

Among the barriers and motivations identified by this review, many are commonly identified as determinants of lifestyle/behavior changes among older adults. However, a few factors are specific to the study /discussed population. Older adults, prominent persons with MCI/dementia, often experience co-morbid conditions and functional limitations that make initiating and maintaining regular exercise challenging [56]. To enable older adults with or at risk for dementia to be physically active, it is essential to both identify and reduce barriers and

augment motivations for exercise. Most barriers to PA and exercise that exist broadly among older adults, in general, are likely to be relevant to persons with MCI/dementia, often to a greater degree [57]. Furthermore, progressive cognitive decline may exacerbate previously existing barriers over time.

While intrinsic factors play a pivotal role, it is essential to acknowledge that extrinsic factors can significantly influence behavior change. For instance, changes in environmental factors, such as accessible facilities and community programs, can encourage older adults with MCI or frailty to engage in PA [58]. Additionally, an individual's socioeconomic status and cultural background may intersect with intrinsic motivations, underscoring the need for tailored interventions that consider diverse contexts [59]. Recognising the correlation between intrinsic and extrinsic factors provides a comprehensive framework for understanding and addressing lifestyle change barriers within these populations.

The intrinsic motivation factors can also be intensified by social support and intervention components. For example, social support increased adherence to exercise, including supervision and encouragement from service providers and carers. This helped exercises become a habitual routine for older adults with MCI [60, 61]. Family dynamics and caregiver support play a crucial role in fostering motivation and overcoming barriers among older adults with MCI and frailty. Integrating a family or carer input into intervention strategies, healthcare professionals can enhance the effectiveness and sustainability of lifestyle interventions in this population [62].

Other than motivation and barriers, we were able to collate input on the preferences towards lifestyle interventions in older adults with frailty and MCI. Specifically tailored interventions based on physical or cognitive ability levels were the most mentioned criteria. Little attention was paid to individual differences and their preferences. Individualized intervention specific to their need and abilities, grouped activities, and cooperating intervention into daily activities were mentioned as their preferences. Intervention in the form of social engagement induced feelings of comfort, enjoyment, and a sense of self-worth among older adults with MCI. Involving them in the co-designing of intervention programs ensures that their preferences and needs are prioritised, fostering a sense of empowerment and ownership over their health [63].

Service provider factors were mentioned in both barriers and preferences themes. Expanding on that, it is important to consider the role of healthcare providers and systems factors in shaping the experiences of older adults with MCI and frailty. For example, the availability and accessibility of geriatric care services, including specialized rehabilitation programs and interdisciplinary care teams, can significantly influence an individual's ability to engage in lifestyle modifications [64]. Moreover, policies related to reimbursement for preventive services and community-based support programs may impact the implementation and sustainability of lifestyle interventions [65]. By addressing the systemic barriers and advocating for policy changes, healthcare professionals can create a more supportive environment for older adults to adopt healthier lifestyles despite cognitive and physical challenges.

The principal strength of this study is the systematic compilation of qualitative studies to extract input on older adults' perceptions of lifestyle interventions. Qualitative methods are ideal for getting an in-depth understanding of related issues. We acknowledge the included articles are not abundant, even after a comprehensive and robust article search. We believe this is due to our aim to specifically look at factors related to lifestyle-changing behavior in a multidomain lifestyle intervention for older adults with MCI and frailty. The limitation of this study is that it includes studies from different countries with various social and environmental differences in which may be different with our local context. Other than that, the severity of

both MCI and the frailty of study participants were not taken into factor while analyzing the findings.

Understanding the unique challenges faced by older adults, particularly those with MCI and frailty, is important for the development of targeted interventions aimed at enhancing their cognitive and physical well-being. The identified intrinsic factors such as existing impairment, lack of knowledge and motivation, exacerbate their cognitive and physical function decline, and highlights the urgency for preventive measures. The intrinsic and extrinsic factors influencing lifestyle modification reported in this review have direct implications for fostering healthy lifestyles among older adults. By recognizing the impact of social support and personalized interventions, our findings may contribute to the broader goal of promoting overall well-being in the ageing population.

The reported preferences for individualized interventions based on physical or cognitive ability levels and the incorporation of activities into daily routines align with initiatives aimed at encouraging sustained lifestyle changes. This holistic understanding is particularly relevant to ongoing efforts in public health to design and implement interventions that not only prevent cognitive decline and frailty but also enhance the overall health and quality of life for older adults. As the landscape of preventive healthcare evolves, our research provides valuable insights that can inform the development of comprehensive strategies for promoting healthy ageing and preventing dementia.

## Conclusion

This review provides insight into the motivation, barriers, and preference toward lifestyle changes among older adults with MCI and physical frailty. The most prominent motivator factors could be classified within the intrinsic motivation theme. Therefore, adopting lifestyle changes should be promoted to generally all older adults, more so to those with impairments such as frailty and MCI.

## Supporting information

**S1 Checklist. Human participants research checklist.**
(DOCX)

**S2 Checklist. Preferred Reporting Items for Systematic Reviews and Meta-Analyses extension for Scoping Reviews (PRISMA-ScR) checklist.**
(DOCX)

## Author Contributions

**Conceptualization:** Suzana Shahar.

**Data curation:** Jamilah Mohammad Hanipah.

**Formal analysis:** Jamilah Mohammad Hanipah, Arimi Fitri Mat Ludin.

**Funding acquisition:** Suzana Shahar.

**Investigation:** Jamilah Mohammad Hanipah.

**Methodology:** Jamilah Mohammad Hanipah, Arimi Fitri Mat Ludin, Devinder Kaur Ajit Singh, Ponnusamy Subramaniam.

**Supervision:** Arimi Fitri Mat Ludin, Devinder Kaur Ajit Singh, Ponnusamy Subramaniam, Suzana Shahar.

**Validation:** Arimi Fitri Mat Ludin, Devinder Kaur Ajit Singh, Ponnusamy Subramaniam.

**Writing – original draft:** Jamilah Mohammad Hanipah.

**Writing – review & editing:** Arimi Fitri Mat Ludin, Devinder Kaur Ajit Singh, Ponnusamy Subramaniam, Suzana Shahar.

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
