## [Decision Letter · Decision Letter 0]

27 Aug 2024

PONE-D-24-29738Motivation, Barriers, and Preferences of Lifestyle Changes among Older Adults with Frailty and Mild Cognitive Impairment: A Scoping Review of Qualitative AnalysisPLOS ONE

Dear Dr. Mat Ludin,

Thank you for submitting your manuscript to PLOS ONE. After careful consideration, we feel that it has merit but does not fully meet PLOS ONE’s publication criteria as it currently stands. Therefore, we invite you to submit a revised version of the manuscript that addresses the points raised during the review process.

We look forward to receiving your revised manuscript.

Kind regards,

Mario Ulises Pérez-Zepeda, M.D., Ph.D.

Academic Editor

PLOS ONE

Journal Requirements:

1. When submitting your revision, we need you to address these additional requirements. Please ensure that your manuscript meets PLOS ONE's style requirements, including those for file naming. The PLOS ONE style templates can be found at https://journals.plos.org/plosone/s/file?id=wjVg/PLOSOne_formatting_sample_main_body.pdf and https://journals.plos.org/plosone/s/file?id=ba62/PLOSOne_formatting_sample_title_authors_affiliations.pdf 2. We note that the grant information you provided in the ‘Funding Information’ and ‘Financial Disclosure’ sections do not match.  When you resubmit, please ensure that you provide the correct grant numbers for the awards you received for your study in the ‘Funding Information’ section. 3. Thank you for stating the following financial disclosure: "This study is funded by the Ministry of Higher Education under the Long-Term Research Grant Scheme (LRGS/1/2019/UM-UKM/1/4)) and approved by the Research Ethics Committee National University of Malaysia (UKM/PPI/111/8/JEP-2020-34)." Please state what role the funders took in the study.  If the funders had no role, please state: "The funders had no role in study design, data collection and analysis, decision to publish, or preparation of the manuscript." If this statement is not correct you must amend it as needed. Please include this amended Role of Funder statement in your cover letter; we will change the online submission form on your behalf. 4. If any supporting files for review show as item type ‘other’ please change to item type ‘supporting info’ as the reviewer does not have access to these ’other’ files. 5. We note that your Data Availability Statement is currently as follows:"All relevant data are within the manuscript and its Supporting Information files." Please confirm at this time whether or not your submission contains all raw data required to replicate the results of your study. Authors must share the “minimal data set” for their submission. PLOS defines the minimal data set to consist of the data required to replicate all study findings reported in the article, as well as related metadata and methods (https://journals.plos.org/plosone/s/data-availability#loc-minimal-data-set-definition). For example, authors should submit the following data: - The values behind the means, standard deviations and other measures reported;- The values used to build graphs;- The points extracted from images for analysis. Authors do not need to submit their entire data set if only a portion of the data was used in the reported study. If your submission does not contain these data, please either upload them as Supporting Information files or deposit them to a stable, public repository and provide us with the relevant URLs, DOIs, or accession numbers. For a list of recommended repositories, please see https://journals.plos.org/plosone/s/recommended-repositories. If there are ethical or legal restrictions on sharing a de-identified data set, please explain them in detail (e.g., data contain potentially sensitive information, data are owned by a third-party organization, etc.) and who has imposed them (e.g., an ethics committee). Please also provide contact information for a data access committee, ethics committee, or other institutional body to which data requests may be sent. If data are owned by a third party, please indicate how others may request data access.

Reviewers' comments:

Reviewer's Responses to Questions

**Comments to the Author**

1. Is the manuscript technically sound, and do the data support the conclusions?

Reviewer #1: Partly

2. Has the statistical analysis been performed appropriately and rigorously? 

Reviewer #1: No

3. Have the authors made all data underlying the findings in their manuscript fully available?

Reviewer #1: No

4. Is the manuscript presented in an intelligible fashion and written in standard English?

Reviewer #1: Yes

5. Review Comments to the Author

Reviewer #1: Introduction

It is necessary to better problematize the relevance of analyzing the effect of lifestyles on physical frailty and cognitive impairment in older people and the relevance of the review on the barriers and motivations of older people to participate in interventions. It would be necessary to define what is understood as lifestyle and lifestyle intervention. A better problematization is necessary and the ultimate goal is to identify the evidence on this topic.

Methodology

-Regarding the methodological proposal of the review, I think it is an interesting proposal with a good structure.

-Regarding stage 3: regarding the inclusion criteria, it is necessary to include whether the year of publication was used as a criterion.

-It is important to describe the scope and limitation of the use of Rayyan Ai software for the selection of evidence. -----Explain better what tasks the software performs, I am concerned that the review is biased by the software and is not a review carried out by the authors.

Review the resolution of Figure 1. It is very blurry. Mention what other sources were used and mentioned in Figure 1. (n=245).

-It would be necessary to add in Figure 1, the number of the study that are interventions but do not find the information on motivations and barriers.

-I believe that the review of lifestyle interventions related to exercise, habits and social participation should be limited to not considering the cognitive dimension or only considering the cognitive dimension because they are two different aspects.

Results, Discussion and Conclusions

-The selection of interventions is not clear, since there are some studies that are not interventions as such and are considered as interventions, such as cognitive impairment studies (Bechard, et al. 2020; Anna Rosenberg et al., 2020, Haesner et al, 2015). The sample size would need to be added in some studies (Bechard).

-The proposal of concepts for the analysis of evidence seems unclear because it does not clearly refer to "motivations" but rather to characteristics or conditions that favor the intervention.

-The proposal for the analysis of evidence lacks support because concepts such as motivations or conditions that favor the intervention are confused. Finally, motivation has to do with something individual that these studies do not seem to show and aspects are being sought that these studies do not have. It is necessary to better work on the objectives of the search and limit it to lifestyle or cognitive interventions. Not to both because it is confusing. It is necessary to be clear about what is meant by motivations or to leave them as scopes or conditions that favor the intervention but not as motivations. If we stick with motivations, it is necessary to better explain the concepts and what they are going to refer to in the analysis of the evidence, it is not very clear.

6. PLOS authors have the option to publish the peer review history of their article (what does this mean?). If published, this will include your full peer review and any attached files.

Reviewer #1: No

---

## [Author Response · Author response to Decision Letter 0]

25 Oct 2024

Thank you for your consideration in reviewing this manuscript and the comments given are highly appreciated to improve the writing.

The response to all the specific comments has been attached as 'Response to Reviewer" document with this submission. 

Thank you.

---

## [Editor Report · Decision Letter 1]

6 Nov 2024

Motivation, barriers and preferences of lifestyle changes among older adults with frailty and mild cognitive impairments: A scoping review of qualitative analysis

PONE-D-24-29738R1

Dear Dr. Mat Ludin,

We’re pleased to inform you that your manuscript has been judged scientifically suitable for publication and will be formally accepted for publication once it meets all outstanding technical requirements.

Kind regards,

Mario Ulises Pérez-Zepeda, M.D., Ph.D.

Academic Editor

PLOS ONE
---

## [Editor Report · Acceptance letter]

12 Nov 2024

PONE-D-24-29738R1 

PLOS ONE

Dear Dr. Mat Ludin, 

I'm pleased to inform you that your manuscript has been deemed suitable for publication in PLOS ONE. Congratulations! Your manuscript is now being handed over to our production team.

Kind regards, 

on behalf of

Dr. Mario Ulises Pérez-Zepeda 

Academic Editor

PLOS ONE